# Is prevalence of e-cigarette and nicotine replacement therapy use among smokers associated with average cigarette consumption in England? A time-series analysis

Emma Beard,[1,2] Jamie Brown,[1,2] Susan Michie,[2] Robert West[1]

[1]Cancer Research UK Health Behaviour Research Centre, University College London, London, UK
[2]Research Department of Educational, Clinical and Health Psychology, University College London, London, UK

**Correspondence to**
Dr Emma Beard;
e.beard@ucl.ac.uk

## ABSTRACT

**Objectives** Many smokers use e-cigarettes and licensed nicotine replacement therapy (NRT), often in an attempt to reduce their cigarette consumption. We estimated how far changes in prevalence of e-cigarette and NRT use while smoking were accompanied by changes in cigarette consumption at the population level.

**Design** Repeated representative cross-sectional population surveys of adults aged 16+ years in England.

**Methods** We used Autoregressive Integrated Moving Average with Exogenous Input (ARIMAX) modelling of monthly data between 2006 and 2016 from the Smoking Toolkit Study. Prevalence of e-cigarette use and NRT use in current smokers, and specifically for smoking reduction and temporary abstinence, were input variables. Mean daily cigarette consumption was the dependent variable. Analyses involved adjustment for mass media expenditure and tobacco-control policies.

**Results** No statistically significant associations were found between changes in use of e-cigarettes (β −0.012, 95% CI −0.026 to 0.002) or NRT (β 0.015, 95% CI −0.026 to 0.055) while smoking and daily cigarette consumption. Neither did we find clear evidence for an association between e-cigarette use (β −0.010, 95% CI −0.025 to 0.005 and β 0.011, 95%−0.027 to 0.004) or NRT use (β 0.006, 95%−0.030 to 0.043 and β 0.022, 95%−0.020 to 0.063) specifically for smoking reduction and temporary abstinence, respectively, and changes in daily cigarette consumption.

**Conclusion** If use of e-cigarettes and licensed NRT while smoking acted to reduce cigarette consumption in England between 2006 and 2016, the effect was likely very small at a population level.

## INTRODUCTION

Randomised controlled trials have shown that use of non-tobacco nicotine-containing products (eg, nicotine replacement therapy; NRT) are efficacious for harm-reduction attempts.[1] Harm reduction is defined as any attempt to reduce the harm from smoking without an intention to quit completely, such as, the use of NRT for smoking reduction (ie, during

## Strengths and limitations of this study

► This is the first time series study to assess the population-level impact of the use of nicotine replacement therapy and e-cigarettes for harm reduction on cigarette consumption.
► This study uses a large representative sample of the population in England and considers both smoking reduction and temporary abstinence.
► A wide range of confounders are adjusted for including population-level interventions.
► In countries with weaker tobacco control, or stricter regulation of using products for harm reduction, different effects may be observed.
► Data are observational and so strong conclusions regarding cause and effect cannot be made.

attempts to cut down) or during periods of temporary abstinence (ie, during periods of time when one is unable to smoke).[1] Outside of the clinical setting where little behavioural support is provided, the use of NRT during attempts to cut down smoking appears to increase smoker's propensity to quit, but does not result in significantly large reductions in cigarette consumption.[2–4] Explanations for this include the lack of behavioural support and possible poor compliance with the medical regimen.[5 6]

In recent years, there has been an increase in the overall use of nicotine-containing products for harm reduction, with a growth in e-cigarettes more than offsetting a decline in the use of NRT.[7–9] Previous studies suggest that e-cigarettes which contain nicotine reduce cravings more effectively than NRT,[7 10 11] have better adherence rates[7 12] and deliver clinically significant levels of nicotine into the blood, at least for some smokers.[10 11 13] Thus, although further studies are needed it is possible that e-cigarettes may be a more

effective aid for smoking reduction than licensed nicotine products.[14 15] However, it also remains possible that e-cigarettes will not result in clinically significant reductions in cigarette intake at a population level.

The aim of this study was to assess the association between changes in prevalence of e-cigarettes and NRT with changes in mean cigarette consumption per day using a time-series approach. Time-series analysis allows us to take into account underlying trends, the effect of other tobacco-control interventions, autocorrelation (whereby data collected at points closer in time tend to be more similar), and to consider possible lag effects of the independent variable on the dependent variable.[16] Where associations are found, they cannot unequivocally establish a causal association but can be indicative, as has been the case with estimating the effect of price of cigarettes on population consumption,[17] mass-media expenditure on use of specialist stop-smoking services[18] and introduction of varenicline to the market on prevalence of use of smoking cessation medication.[19] Where associations are not found, or they go in a direction opposite to that expected, this can also be informative.

Specifically, this paper assesses the association between mean cigarette consumption per day and:
1. Current e-cigarette use among smokers for any purpose, current use specifically for smoking reduction and current use specifically for temporary abstinence.
2. Current NRT use among smokers for any purpose, current use specifically for smoking reduction and current use specifically for temporary abstinence.

Sensitivity analyses will examine the effect of focusing only on daily e-cigarette and NRT use, given previous associations between extent of non-tobacco nicotine-containing product use and the effectiveness of harm-reduction attempts.[6]

## METHODS
### Design
We used Autoregressive Integrated Moving Average with Exogeneous Input (ARIMAX) modelling of monthly data between 2006 and 2016 primarily from the Smoking Toolkit Study. The smoking toolkit study (STS) is a monthly survey of a representative sample of the population in England aged 16+ years.[20] This has been collecting data on smoking patterns among smokers and recent ex-smokers since November 2006. Questions on the use of e-cigarettes among all smokers were introduced in May 2011 and as aids to a quit attempt among smokers attempting to stop in July 2009. The STS involves monthly household surveys using a random location sampling design, with initial random selection of grouped output areas (containing 300 households), stratified by ACORN (sociodemographic) characteristics (https://acorn.caci.co.uk/) and region. Interviewers then choose which houses within these areas are most likely to fulfil quotas based on the probability of individuals being at home in different regions and conduct face-to-face computer-assisted interviews with one member per household. Participants from the STS appear to be representative of the population in England, having similar sociodemographic composition as other large national surveys, such as the Health Survey for England.[20]

### Measures
#### Explanatory variables
Daily and non-daily smokers were asked the following questions:
1. Which, if any, of the following are you currently using to help you cut down the amount you smoke?
2. Do you regularly use any of the following in situations when you are not allowed to smoke?
3. Can I check, are you using any of the following either to help you stop smoking, to help you cut down or for any other reason at all?

All three questions had the following response options: nicotine gum, nicotine replacement lozenges\tablets, nicotine replacement inhaler, nicotine replacement nasal spray, nicotine patch, electronic cigarette, nicotine mouth spray, other, none.

Current e-cigarette use was derived by an 'electronic cigarette' response to any of the three questions; e-cigarette use for smoking reduction by a response to the first question; and e-cigarette use for temporary abstinence by a response to the second question.

Current NRT use was derived by an NRT product response ('nicotine gum, nicotine replacement lozenges\tablets, nicotine replacement inhaler, nicotine replacement nasal spray, nicotine patch or nicotine mouth spray') to any of the three questions; NRT use for smoking reduction by an NRT product response to the first question; and NRT use for temporary abstinence by an NRT product response to the second question.

Data were not recorded on NRT use for temporary abstinence between November 2006 and January 2007 and was imputed using prevalence data from February 2007.

Data were only available on the prevalence of use of electronic cigarettes among smokers from April 2011 although use specifically during a recent quit attempt were available from July 2009. Thus, prevalence of electronic cigarette use among smokers between July 2009 and April 2011 was estimated from data on use during a quit attempt; use of electronic cigarettes among smokers between November 2006 and June 2009 was assumed to be 0.1% of smokers based on other surveys which found their use to be very rare before 2009.[21 22]

Daily NRT and e-cigarette users were classified as those who reported that they used the product(s) at least once per day in response to the question: How many times per day on average do you use your nicotine replacement product or products? This question was introduced in July 2010. Prior to this time, prevalence of daily NRT use was assumed to be 60% of all users,[6] while e-cigarette prevalence was computed as above using prevalence during a quit attempt or 0.1%.

## Outcome variables

Smokers taking part in the STS were also asked how many cigarettes they smoke on average per day. Non-daily smokers were asked how many cigarettes they smoked per week which was then converted to a daily figure.

## Co-variables

In England, tobacco mass media campaigns have been run as part of a national tobacco-control programme. Spending was almost completely suspended in 2010 and then reintroduced in 2011 at a much lower level. Previous studies have shown that such cuts were associated with a decreased use of smoking cessation support.[18 23] Thus, advertising expenditure will be adjusted for using data obtained from Public Health England. Data on mass media expenditure was available monthly from May 2008, and yearly prior to this period, and so a monthly average was assumed. For a number of months, spending was effectively zero and was imputed as 0.1 to allow the analysis to run.

A number of tobacco-control policies were adjusted for. These included the move in commissioning of stop-smoking services to local authorities in April 2013,[24] introduction of a smoking ban in July 2007,[25] licensing of NRT for harm reduction in December 2009,[26] the publication of National Institute for Health and Care Excellence guidance on harm reduction in June 2013[27] and change in the minimum age of sale of cigarettes in October 2007.[28] Price of cigarettes is correlated 0.99 with time and will thereby be taken into account by use of differencing (ie, using the differences between consecutive observation rather than observations themselves) to make the series stationary.

## Analysis

The analysis plan was registered on the Open Science Framework prior to data analysis (https://osf.io/6swk3/). All data were analysed in R V.3.2.4[29] using ARIMAX modelling.[16 30 31] Data were weighted prior to the analyse to match the population in England using a rim (marginal) weighting technique. This involves an iterative sequence of weighting adjustments whereby separate nationally representative target profiles are set (for gender, working status, children in the household, age, social grade and region). This process is then repeated until all variables match the specified targets.[20]

Two waves of data were collected in March 2007 and March 2013. These waves were averaged. No data were collected in December 2008. Mean cigarette consumption, NRT use and e-cigarette use during this period were calculated as an average of the month before and the month after. For a few months (May 2012, July 2012, September 2012, November 2012, January 2013, March 2013), data on electronic cigarettes and NRT use among smokers were not recorded. For these months, the average of the previous and next month was imputed.

The Granger causality test suggested that there was some evidence for the violation of the assumption of weak exogeneity (ie, Y can depend on the lagged values of X but the reverse must not be true) between the input and the output series. However, caution has been advised when using this and similar tests on data across a long time series,[32 33] and there was no theoretical reason we could identify for a bidirectional relationship between e-cigarette use and cigarette consumption. It was assumed that the association was spurious and likely removed following adjustment for other covariates.

Both unadjusted and fully adjusted models are reported which regressed onto mean cigarette consumption per day: (1) use of e-cigarettes among current smokers; (2) use of e-cigarettes for smoking reduction; (3) use of e-cigarettes for temporary abstinence; (4) use of NRT for harm reduction; (5) use of NRT for temporary abstinence and (6) use of NRT for smoking reduction. Sensitivity analyses were conducted which constrained the analysis to only those reporting daily e-cigarette and NRT use. We followed a standard ARIMAX modelling approach.[16 34] The series were first log-transformed to stabilise the variance, and if required, first differenced and seasonally differenced. The autocorrelation and partial autocorrelation functions were then examined in order to determine the seasonal and non-seasonal moving average (MA) and autoregressive terms (AR). For example, AR(1) means that the value of a series at one point in time is the sum of a fraction of the value of the series at the immediately preceding point in time and an error component; while MA(1) means that the value of a series at one point in time is a function of a fraction of the error component of the series at the immediately preceding point in time and an error component at the current point in time. To identify the most appropriate transfer function (ie, lag) for the continuous explanatory variables, the sample cross-correlation function was checked for each ARIMAX model. Coefficients can be interpreted as estimates of the percentage change in cigarette consumption for every (a) percentage increase in use of e-cigarettes and NRT, (b) percentage increase in mass media expenditure and (c) implementation of tobacco-control policies.

Bayes factors (BFs) were derived for non-significant findings using an online calculator[35] to disentangle whether there is evidence for the null hypothesis of no effect (BF <1/3rd) or the data are insensitive (BF between 1/3rd and 3). A half-normal distribution was assumed with a percentage change in the outcomes of interest for every percentage increase in the input series of 0.009% based on the effect detectable with 80% power (see sample size). Sensitivity analyses were conducted using a much larger percentage change of 0.1. This was based on a meta-analysis assessing the efficacy of non-to-bacco nicotine replacement products for harm reduction which reported that 21.8% of the experimental group had reduced consumption by more than 50% at final follow-up compared with 16.5% receiving placebo.[1] We therefore assumed that a 5% change in prevalence of NRT and e-cigarettes would be associated with a 0.5% change in overall cigarette consumption.

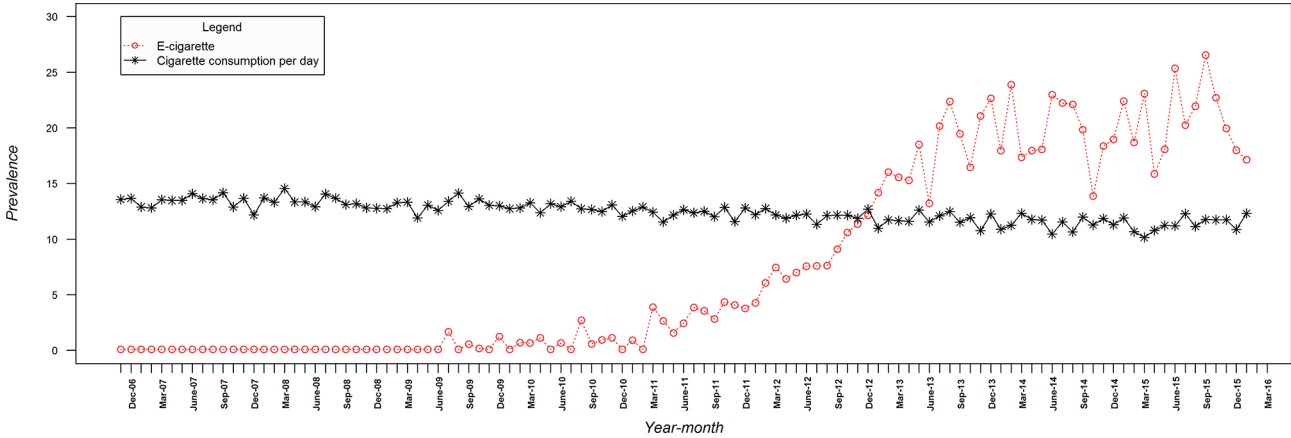

**Figure 1** Monthly prevalence of cigarette consumption and e-cigarettes for harm reduction among smokers.

Strengthening the Reporting of Observational Studies in Epidemiology guidelines for the reporting of observational studies were followed throughout.[36]

### Sample size

Simulation-based power analyses suggested that this study would have 80% power to detect a change in the output series of 0.009% for every 1% change in the input series, assuming 113 monthly data collection points, MA (1) autocorrelation,[37] a baseline proportion for the input series of 0.005,[9] a baseline mean (SD) for the output series of 12.3[38] and a total change over time for the input series of 30%.[38]

### RESULTS

#### Sample characteristics

Data were collected on 199 483 adults aged 16+ years taking part in the STS who reported their smoking status between November 2006 and March 2016. Of these, 43 608 (20.8%, 95% CI 20.6 to 21.0) were current smokers. Fifty-two per cent (95% CI 52% to 53%) of the smokers were male and 60.4% (95% CI 60% to 60.1%) were in routine or manual positions or were unemployed.

The average age of smokers in this study was 42.1 years (95% CI 42.0 to 42.1).

### Main analysis

Figure 1 shows that cigarette consumption declined over the study period from 13.6 to 12.3 (mean 12.4, SD 0.92). This figure also shows that current use of e-cigarettes among smokers for harm reduction increased from negligible use in the last quarter of 2006 to 17.1% at the end of the study (mean 7.8%, SD 8.82). Figure 2 shows that there was also a decline in the use of NRT for harm reduction from 12.2% to 6% (mean 14.4%, SD 4.36). Online supplementary figures 1 and 2 show the changes in e-cigarette and NRT use for smoking reduction and temporary abstinence, respectively.

Tables 1, 2 and 3 show the results of the ARIMAX models assessing the association between cigarette consumption per day with (1) e-cigarette use among current smokers and NRT use for harm reduction; (2) e-cigarette and NRT use for smoking reduction and (3) e-cigarette and NRT use for temporary abstinence. The findings were inconclusive as to whether an association was present between use of e-cigarettes and NRT for any purpose and cigarette consumption.

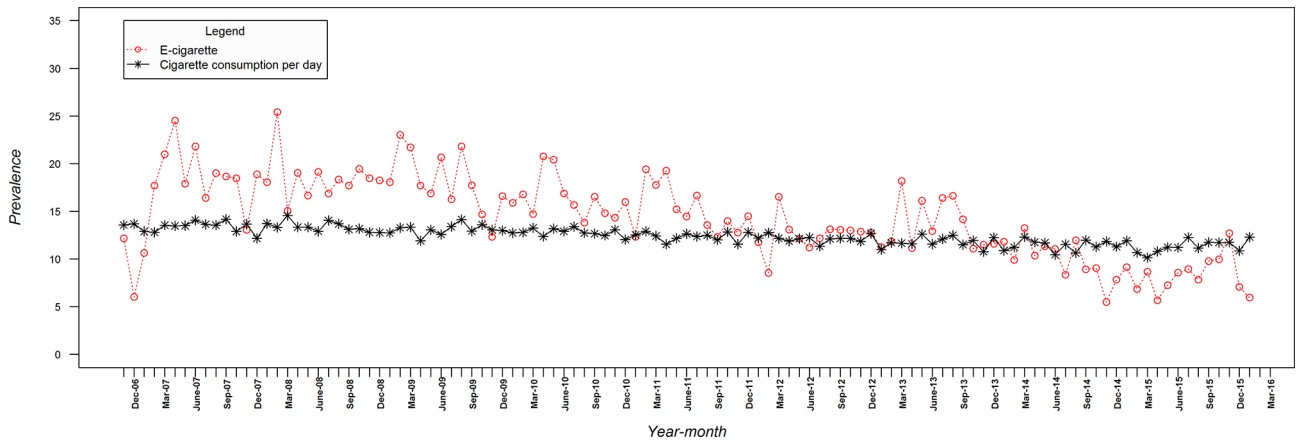

**Figure 2** Monthly prevalence of cigarette consumption and nicotine replacement therapy use for harm reduction among smokers.

**Table 1** Estimated percentage-point changes in mean cigarette consumption per day as a function of e-cigarette use and NRT use among smokers from November 2006 to March 2016, based on ARIMAX models

| | All users of nicotine replacement | | Only daily users of nicotine replacement | |
|---|---|---|---|---|
| | Percentage change per 1 % change in the exposure (95% CI) P values | Total percentage change due to the exposure (95% CI) P values | Percentage change per 1 % change in the exposure (95% CI) P values | Total percentage change due to the exposure (95% CI) P values |
| Any current use of e-cigarettes (immediate impact) | −0.012 (−0.026 to 0.002) 0.091 | −0.011 (−0.025 to 0.002) 0.097 | −0.010 (−0.024 to 0.004) 0.149 | −0.011 (−0.026 to 0.003) 0.130 |
| NRT use for harm reduction (immediate impact) | 0.015 (−0.026 to 0.055) 0.475 | 0.012 (−0.028 to 0.053) 0.546 | 0.003 (−0.019 to 0.025) 0.794 | 0.005 (−0.017 to 0.027) 0.672 |
| Mass media expenditure (immediate impact) | <0.001 (−0.001 to 0.001) 0.984 | <0.001 (−0.001 to 0.001) | <0.001 (−0.001 to 0.001) | <0.001 (−0.001 to 0.001) 0.880 |
| Smoking ban (pulse effect) | | 0.015 (−0.070 to 0.101) 0.724 | | 0.013 (−0.072 to 0.099) 0.756 |
| Increase in age-of-sale (pulse effect) | | −0.041 (−0.126 to 0.044) 0.342 | | −0.043 (−0.128 to 0.042) 0.324 |
| Move to local authority control (pulse effect) | | −0.019 (−0.105 to 0.067) 0.662 | | −0.027 (−0.112 to 0.058) 0.533 |
| Licensing for NRT for harm reduction (pulse effect) | | 0.021 (−0.067 to 0.110) 0.639 | | 0.020 (−0.069 to 0.109) 0.661 |
| NICE guidance on harm reduction (pulse effect) | | −0.024 (−0.109 to 0.061) 0.578 | | −0.028 (−0.114 to 0.057) 0.512 |
| Best fitting model | ARIMAX(0,1,1)(0,0,0)[12] | ARIMAX(0,1,1)(0,0,0)[12] | ARIMAX(0,1,1)(0,0,0)[12] | ARIMAX(0,1,1)(0,0,0)[12] |
| Non-seasonal AR p value | NA | NA | NA | NA |
| Non-seasonal MA p value | <0.001 | <0.001 | <0.001 | <0.001 |

Continued

**Table 1** Continued

| | All users of nicotine replacement | | Only daily users of nicotine replacement | | |
| --- | --- | --- | --- | --- | --- |
| | Percentage change per 1 % change in the exposure (95% CI) P values | | | | |
| Seasonal AR p value | NA | NA | NA | NA | NA |
| Seasonal MA p value | NA | NA | NA | NA | NA |
| $R^2$ | 0.65 | 0.66 | 0.65 | 0.64 | 0.66 |
| Bayes factor e-cigarette (0.009 (0.1)) | 2.44 (0.46) | 2.68 (0.55) | 1.95 (0.35) | | 2.12 (0.41) |
| Bayes factor NRT (0.009 (0.1)) | 0.77 (0.14) | 0.74 (0.13) | 0.69 (0.09) | | 0.63 (0.08) |

An AR(1) means that the value of a series at one point in time is the sum of a fraction of the value of the series at the immediately preceding point in time and an error component; an MA(1) means that the value of a series at one point in time is a function of a fraction of the error component of the series at the immediately preceding point in time and an error component at the current point in time.

AR, autoregressive; ARIMAX, Autoregressive Integrated Moving Average with Exogeneous Input; MA, moving average; NA, not applicable; NICE, National Institute for Health and Care Excellence; NRT, nicotine replacement therapy.

BFs were between one-third and three when assuming a 0.009% change in cigarette consumption for every percentage change in the input series, suggesting the data are insensitive to detect very small reductions in cigarette consumption. Most BFs were less than one-third, when assuming a 0.1% change in cigarette consumption for every percentage change in the input series, suggesting evidence for the null hypothesis that NRT use and e-cigarette use among smokers has not resulted in large reductions in cigarette intake.

### Sensitivity analysis

Current daily use of e-cigarettes among smokers for harm reduction increased from negligible use in the last quarter of 2006 to 11.1% at the end of the study (mean 4.5%, SD 4.91). There was also an increase in e-cigarette use specifically for temporary abstinence (from 0.1% to 8.4%; mean 3.5% SD 3.81) and smoking reduction (from 0.1% to 8.3%; mean 3.3% SD 3.64).

In contrast, there was a decline in the use of NRT for harm reduction from 7.3% to 2.9% (mean 6.5%, SD 2.35) and a decline in NRT use specifically for temporary abstinence (from 7.3% to 1.8%; mean 4.7% SD 2.29) and smoking reduction (from 6.8% to 2.6%; mean 5.8%, SD 2.46).

Tables 1, 2 and 3 also show the results of the sensitivity analyses restricted to those smokers using NRT or e-cigarettes daily. The findings were inconclusive as to whether or not an association was present between the daily use of e-cigarettes and NRT for any purpose and cigarette consumption. BFs suggested the data are insensitive to detect very small reductions in cigarette consumption, but there is evidence for the null hypothesis that NRT use and e-cigarette use among smokers have not resulted in large reductions in cigarette intake.

## DISCUSSION

To our knowledge, this is the first empirical study to estimate the population association between the use of e-cigarettes and NRT among current smokers on cigarette consumption per day, using a time-series approach. There was evidence that there was no substantial association between the rise in use of e-cigarettes and decline in NRT use and changes in cigarette consumption per day.

### Strengths and limitations

A strength of the study is the use of a large representative sample of the population in England, stratification of results by daily use, and the consideration of both temporary abstinence and smoking reduction. Previous studies have shown that reductions in cigarette intake are dependent on the extent of NRT use and differ as a function of the specific harm-reduction behaviour, that is, an attempt to cut down or restraining from smoking during periods of brief abstinence.[2 6]

The study had a number of limitations. First, caution should be taken when interpreting estimates of the

**Table 2** Estimated percentage point changes in mean cigarette consumption per day as a function of e-cigarette use and NRT use among smokers for cutting down from November 2006 to March 2016, based on ARIMAX models

| | All users of nicotine replacement | | Only daily users of nicotine replacement | |
|---|---|---|---|---|
| | Percentage change per 1 % change in the exposure (95% CI) P values | Total percentage change due to the exposure (95% CI) P values | Percentage change per 1 % change in the exposure (95% CI) P values | Total percentage change due to the exposure (95% CI) P values |
| Use of e-cigarettes for cutting down (immediate impact) | −0.010 (−0.024 to 0.005) 0.191 | −0.010 (−0.025 to 0.005) 0.191 | −0.008 (−0.023 to 0.006) 0.256 | −0.009 (−0.024 to 0.006) 0.229 |
| NRT use for cutting down (immediate impact) | 0.002 (−0.033 to 0.037) 0.917 | 0.006 (−0.030 to 0.043) 0.732 | −0.002 (−0.016 to 0.013) 0.825 | −0.002 (−0.017 to 0.013) 0.786 |
| Mass media expenditure (immediate impact) | <0.001 (−0.001 to 0.001) 0.885 | <0.001 (−0.001 to 0.001) 0.885 | <0.001 (−0.001 to 0.001) | <0.001 (−0.001 to 0.001) 0.860 |
| Smoking ban (pulse effect) | | 0.014 (−0.072 to 0.099) 0.755 | | 0.012 (−0.073 to 0.097) 0.782 |
| Increase in age-of-sale (pulse effect) | | −0.043 (−0.128 to 0.042) 0.323 | | −0.042 (−0.127 to 0.043) 0.329 |
| Move to local authority control (pulse effect) | | −0.025 (−0.110 to 0.061) 0.571 | | −0.029 (−0.115 to 0.056) 0.499 |
| Licensing for NRT for harm reduction (pulse effect) | | 0.018 (−0.072 to 0.108) 0.694 | | 0.015 (−0.074 to 0.103) 0.747 |
| NICE guidance on harm reduction (pulse effect) | | −0.028 (0.058 to <0.001) 0.529 | | −0.027 (−0.112 to 0.059) 0.541 |
| Best fitting model | ARIMAX(0,1,1)(0,0,0)[12] | ARIMAX(0,1,1)(0,0,0)[12] | ARIMAX(0,1,1)(0,0,0)[12] | ARIMAX(0,1,1)(0,0,0)[12] |
| Non-seasonal AR p values | NA | NA | NA | NA |
| Non-seasonal MA p values | <0.001 | <0.001 | <0.001 | <0.001 |

Continued

**Table 2** Continued

| | All users of nicotine replacement | | Only daily users of nicotine replacement | |
|---|---|---|---|---|
| | Percentage change per 1 % change in the exposure (95% CI) | P values | Percentage change per 1 % change in the exposure (95% CI) | P values |
| Seasonal AR p values | NA | NA | NA | NA |
| Seasonal MA p values | NA | NA | NA | NA |
| $R^2$ | 0.64 | 0.65 | 0.64 | 0.65 |
| Bayes factor e-cigarette (0.009 (0.1)) | 1.87 (0.34) | 1.79 (0.32) | 1.46 (0.23) | 1.61 (0.27) |
| Bayes factor NRT (0.009 (0.1)) | 0.86 (0.16) | 0.81 (0.15) | 0.76 (0.10) | 0.76 (0.10) |

An AR(1) means that the value of a series at one point in time is the sum of a fraction of the value of the series at the immediately preceding point in time and an error component; an MA(1) means that the value of a series at one point in time is a function of the error component of the series at the immediately preceding point in time and an error component at the current point in time.

AR, autoregressive; ARIMAX, Autoregressive Integrated Moving Average with Exogeneous Input; MA, moving average; NA, not applicable; NICE, National Institute for Health and Care Excellence; NRT, nicotine replacement therapy.

covariates, that is, impact of some of the tobacco-control policies, as interrupted explanatory variables with short time-periods prior to their introduction in ARIMAX-type models often give inaccurate estimates of the SEs.[28] Thus, although the increase in age-of-sale has been previously associated with a decline in smoking prevalence,[24] the short lead-in period may have masked any true association.[27] Second, the STS required participants to recall their average daily cigarette intake which is likely to have been somewhat inaccurate. Third, the findings may not generalise to other countries. England has a strong tobacco-control climate and relatively liberal attitude towards harm reduction and e-cigarette use. In countries with weaker tobacco control, or stricter regulation of using products for harm reduction, different effects may be observed. Fourth, although we are unaware of any other major population-level interventions or other events during the study period, we cannot rule out residual confounding. Fifth, participants were not asked questions regarding potentially important features of the e-cigarette (eg, nicotine content, flavouring, device type) or frequency and duration of use. It is likely that these factors may play a role in their effectiveness and should be considered in future studies.[15 39] Finally, as data were not collected on current e-cigarette use prior to April 2011, prevalence was estimated from use during a quit attempt or from previous studies.[21 22] This was necessary to ensure that the time series was long enough for an ARIMAX analysis and is an appropriate approach when data are missing completely at random.[16 40] As prevalence was low and relatively stable during this period, it is unlikely to have impacted on the reported results.

### Implications of findings

The findings are in line with previous studies which show that reductions in cigarette consumption observed in clinical trials of NRT for harm reduction do not appear to generalise beyond the closely controlled trial setting.[1 2] It was hypothesised that e-cigarettes may be associated with population mean cigarette intake given that they reduce cravings more effectively than NRT,[7 10 11] have better adherence rates[7 12] and deliver clinically significant levels of nicotine into the blood.[10 11 11 13]

The finding that e-cigarette use was not associated with reductions in consumption at a population level is consistent with previous real-world studies at the individual level. These have found little change in consumption among ever e-cigarette users[41] and that only a minority of daily users manage to reduce by a substantial amount which is not likely to be detected at a population level.[42] The findings of a recent pragmatic controlled trial, whereby 60% of participants using e-cigarettes had managed to reduce by over 50% by 6 months' follow-up, suggests that the lack of effectiveness at a population level may not be the consequence of poor behavioural support.[11]

Of course, it remains plausible that e-cigarettes may still be associated with a small effect on mean population cigarette consumption,[15] and that a reduction in harm from

**Table 3** Estimated percentage point changes in mean cigarette consumption per day as a function of e-cigarette use and NRT use among smokers for temporary abstinence from November 2006 to March 2016, based on ARIMAX models

| | All users of nicotine replacement | | Only daily users of nicotine replacement | |
|---|---|---|---|---|
| | Percentage change per 1 % change in the exposure (95 % CI) P values | Total percentage change due to the exposure (95% CI) P values | Percentage change per 1 % change in the exposure (95 % CI) P values | Total percentage change due to the exposure (95% CI) P values |
| Use of e-cigarettes for temporary abstinence (immediate impact) | −0.010 (−0.024 to 0.005) 0.150 | −0.011 (−0.027 to 0.004) 0.146 | −0.010 (−0.024 to 0.004) 0.159 | −0.011 (−0.026 to 0.003) 0.135 |
| NRT use for temporary abstinence (immediate impact) | 0.023 (−0.016 to 0.062) 0.241 | 0.022 (−0.020 to 0.063) 0.303 | 0.006 (−0.015 to 0.028) 0.563 | 0.006 (−0.016 to 0.028) 0.585 |
| Mass media expenditure (immediate impact) | <0.001 (−0.001 to 0.001) 0.873 | <0.001 (−0.001 to 0.001) | <0.001 (−0.001 to 0.001) | <0.001 (−0.001 to 0.001) 0.942 |
| Smoking ban (pulse effect) | | 0.017 (−0.069 to 0.103) 0.696 | | 0.014 (−0.071 to 0.099) 0.750 |
| Increase in age-of-sale (pulse effect) | | −0.036 (−0.122 to 0.050) 0.415 | | −0.040 (−0.125 to 0.044) 0.350 |
| Move to local authority control (pulse effect) | | −0.016 (−0.102 to 0.071) 0.721 | | −0.026 (−0.111 to 0.060) 0.556 |
| Licensing for NRT for harm reduction (pulse effect) | | 0.023 (−0.067 to 0.114) 0.615 | | 0.019 (−0.070 to 0.108) 0.670 |
| NICE guidance on harm reduction (pulse effect) | | −0.021 (−0.106 to 0.065) 0.638 | | −0.030 (−0.116 to 0.055) 0.483 |
| Best fitting model | ARIMAX(0,1,1)(0,0,0)[12] | ARIMAX(0,1,1)(0,0,0)[12] | ARIMAX(0,1,1)(0,0,0)[12] | ARIMAX(0,1,1)(0,0,0)[12] |
| Non-seasonal AR P values | NA | NA | NA | NA |

Continued

**Table 3**  Continued

| | All users of nicotine replacement | | Only daily users of nicotine replacement | |
|---|---|---|---|---|
| | Percentage change per 1 % change in the exposure (95% CI) P values | | | |
| Non-seasonal MA P values | <0.001 | <0.001 | <0.001 | <0.001 |
| Seasonal AR P values | NA | NA | NA | NA |
| Seasonal MA P values | NA | NA | NA | NA |
| $R^2$ | 0.65 | 0.65 | 0.64 | 0.65 |
| Bayes factor e-cigarette (0.009 (0.1)) | 1.01 (0.59) | 1.94 (0.38) | 1.97 (0.35) | 2.15 (0.41) |
| Bayes factor NRT (0.009 (0.1)) | 0.15 (0.02) | 0.69 (0.11) | 1.05 (0.18) | 0.61 (0.08) |

An AR(1) means that the value of a series at one point in time is the sum of a fraction of the value of the series at the immediately preceding point in time and an error component; an MA(1) means that the value of a series at one point in time is a function of a fraction of the error component of the series at the immediately preceding point in time and an error component at the current point in time.

AR, autoregressive; ARIMAX, Autoregressive Integrated Moving Average with Exogeneous Input; MA, moving average; NA, not applicable; NICE, National Institute for Health and Care Excellence; NRT, nicotine replacement therapy.

smoking at a population level could be seen through their promotion of quit attempts[37] or by reducing smoke intake from each cigarette.[5]

## Conclusion

In conclusion, the increased prevalence of e-cigarettes use among smokers in England has not been associated with a detectable change in cigarette consumption per day. The decline in the use of NRT has also not been associated with a change in mean cigarette intake. If use of e-cigarettes and licensed NRT while smoking act to reduce cigarette consumption, the effect is probably small.

**Contributors** EB, JB, SM and RW designed the study. EB wrote the first draft and conducted the analyses. All authors commented on this draft and contributed to the final version.

**Funding** The Smoking Toolkit Study is currently primarily funded by Cancer Research UK (C1417/A14135; C36048/A11654; C44576/A19501) and has previously also been funded by Pfizer, GSK and the Department of Health. JB's post is funded by a fellowship from the Society for the Study of Addiction and CRUK also provides support (C1417/A14135). RW is funded by Cancer Research UK (C1417/A14135). EB is funded by a fellowship from the NIHR SPHR (SPHR-SWP-ALC-WP5) and CRUK also provides support (C1417/A14135). SW is funded by Cancer Research UK (C1417/A14135) and NIHR SPHR (SPHR-SWP-ALC-WP5) also provide support. SPHR is a partnership between the Universities of Sheffield; Bristol; Cambridge; Exeter; UCL; The London School for Hygiene and Tropical Medicine; the LiLaC collaboration between the Universities of Liverpool and Lancaster and Fuse; The Centre for Translational Research in Public Health, a collaboration between Newcastle, Durham, Northumbria, Sunderland and Teesside Universities.

**Disclaimer** The views expressed are those of the authors(s) and not necessarily those of the NHS, NIHR or Department of Health. No funders had any involvement in the design of the study, the analysis or interpretation of the data, the writing of the report or the decision to submit the paper for publication.

**Competing interests** RW undertakes consultancy and research for and receives travel funds and hospitality from manufacturers of smoking cessation medications but does not, and will not take funds from e-cigarettes manufacturers or the tobacco industry. RW and SM are honorary co-directors of the National Centre for Smoking Cessation and Training. RW is a Trustee of the stop-smoking charity, QUIT. RW's salary is funded by Cancer Research UK. SM's salary is funded by Cancer Research UK and by the National Institute for Health Research (NIHR)'s School for Public Health Research (SPHR). EB and JB have received unrestricted research funding from Pfizer. EB and JB are funded by CRUK. EB is also funded by NIHR's SPHR and JB by the Society for the Study of Addiction. RW has received travel funds and hospitality from, and undertaken research and consultancy for pharmaceutical companies that manufacture or research products aimed at helping smokers to stop. These products include nicotine replacement therapies, Champix (varenicline) and Zyban (bupropion). This has led to payments to him personally and to his institution.

**Patient consent** Obtained.

**Ethics approval** Ethical approval for the Smoking Toolkit Study was granted by the UCL Ethics Committee (ID 0498/001).

**Provenance and peer review** Not commissioned; externally peer reviewed.

**Data sharing statement** For access to the data please contact the lead author, EB (e.beard@ucl.ac.uk).

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
