## [Reviewer comments · BMJ Open]

ARTICLE DETAILS

TITLE (PROVISIONAL)	Is prevalence of e-cigarette and nicotine replacement therapy use among smokers associated with average cigarette consumption in England? A time-series
AUTHORS	Beard, Emma; Brown, Jamie; Michie, Susan; West, Robert

VERSION 1 – REVIEW

REVIEWER	Susanna Raisamo National Institute for Health and Welfare, Finland
REVIEW RETURNED	30-Jan-2017

GENERAL COMMENTS	I am happy for a chance to review this interesting manuscript. This is a concise and well written paper. I do not feel adequately qualified to assess the statistics made with R. However, the results are presented clearly and the chosen methods of data analyses seem appropriate. Overall, I think that data and the findings are worthy of publication. I only have few minor comments for possible consideration. I think that the authors missed their opportunity to discuss the findings and their possible implications further in the Discussion. For example, what kind of implication these findings provide e.g. for the current public and/or scientific debate around harm reduction. There are many types of NRT products available. Moreover, there are other forms of tobacco use (snus/snuff). These aspects might be worth to consider in the Discussion as well. In future studies, supplemental analysis taking into account SES would provide interesting results. There could be greater accumulation of e-cigarettes and NRT products use in high-SES than low-SES smokers. Introduction (1st pg): References are mainly limited to the papers made by the author in England. If relevant data is published outside England, I strongly recommend citations of other related work as well. Finally, I wonder how response pattern and non-participation to these surveys might have changed over time. Authors could briefly touch this issue in the manuscript.
--

REVIEWER	Marjolein Verbiest National Institute for Health Innovation School of Population Health The University of Auckland New Zealand
REVIEW RETURNED	16-Mar-2017

GENERAL COMMENTS

Association between e-cigarette and nicotine replacement therapy use among smokers with cigarette consumption in England: a time-series analysis

This paper describes the results of a time series analysis using ARIMAX modelling to evaluate trends in e-cigarettes and NRT use between 2006 and 2016 among tobacco smokers in England. More in particular, the analyses examined the association between e-cigarette and NRT use for different purposes (i.e. harm reduction and temporary abstinence) and overall cigarette consumption. The results show a decrease in overall cigarette consumption, an increase in e-cigarette use and a decrease in the use of NRT. No associations were found between the decline in cigarette consumption and changes in e-cigarette and NRT use.

The ecological design and time series approach are strong features of the study which makes the paper an important addition to the current literature around e-cigarettes. Yet several changes will improve the paper as outlined below.

Abstract

- Design: only the data collection procedures are described. Please add that it concerns a population-based or ecological study design.
- Conclusion: This conclusion is only based on the findings in this study. Please nuance the conclusion by stating: "Based on this study ..." and at a term such as "may" or "is likely to".

Introduction

- Page 3, line 17: Which randomised controlled trials are you referring to, please add references.
- Page 3, line 17: What is meant by 'non-tobacco nicotine containing products'. Please add some examples (i.e. such as...)
- Page 3, line 28: What do you mean with 'licensed products'? This may differ from one country to the other. Please specify.
- Page 3, line 30: There's a distinction between e-cigarettes with and without nicotine. Please specify here that concerns e-cigarette with nicotine.
- Page 3, line 31: Add the Cochrane Review on the effectiveness of e-cigarettes for smoking abstinence
- Page 3, line 32/33: "This, e-cigarettes may be a more effective aid..."; the Cochrane Review concludes that it's not yet possible to determine if e-cigarettes are more effective than NRT in helping people to quit. Please add this to this section.
- Page 3, line 37-49 & page 4 line 3-7: This section describes the methods which are used to examine the hypotheses. This should not be described. Please focus on describing the research question and hypotheses in the last paragraph of the introduction and describe the methods in the methods section.

Methods

- Start methods section with a paragraph describing the design of the study
- Overall the writing of this section can be improved to make it more concise and readable (e.g. page 4, line 28-30 "Participants who reported daily or non-daily smoking were asked the following questions")
- Page 4, line 35: Did the survey make a distinction made between e-cigarettes with a without nicotine?

If not, please describe the implication of this in the discussion section.

- Page 4, line 46-54: the authors describe how smokers were categorised into e-cigarette users and NRT users. What happened when smokers uses both NRT and e-cigarettes?
- Page 4, line 46-54: perhaps the authors could add definitions of “smoking reduction”, “harm reduction” and “temporary abstinence” of clarification
- Page 4, line 53: Why was NRT use for temporary abstinence not recorded between Nov-2006 and Jan-2007?
- Page 4, line 53-57 & page 5, line 3-14: A lot of imputations, estimations and assumptions are described and were applied to the data. Can the authors explain why they have chosen for this approach and why they haven’t decided to focus on the available data from 2011 onwards instead? What are the implications of the imputations/estimations and assumptions - please describe this in the discussion section.
- Page 5, line 17-18: “Smokers taking part in the STS...”; How was the average number of cigarettes smoked per day determined for non-daily smokers?
- Page 5, line 35-38: “Price of cigarettes...”; This sentence will not make a lot of sense for someone who is not familiar with time series analysis. Please explain what is this is needed.
- Page 5, line 40: “ANALYSIS” should be a subheading instead of a main heading.
- Page 5, paragraph analysis: ARIMAX modelling is a complex approach. The authors provide an explanation of the analyses procedures but this is rather complex. Perhaps this section could be simplified, keeping in mind that a lot of readers are not familiar with ARIMAX modelling?
- Add a paragraph regarding the ethical considerations in this study (informed consent, ethical approval, etc.)

Results

- Perhaps the authors can consider adding a table with descriptive statistics of the main variables (e.g. for each data collection wave the % of daily and non-daily smokers, the % of e-cigarette and NRT users for the different purposes, and the average number of cigarettes smoked). This table could also include some sociodemographic characteristics of participants (e.g. age, sex, ethnicity, education)
- Page 5, paragraph “Main analyses”: start with describing the trend in the outcome variable (i.e. average daily cigarette consumption), followed by a description of the trends in the explanatory variables.
- Page 5, paragraph “Main analyses”: where the declining trends in cigarette consumption and NRT use and the increasing trend in e-cigarette use statistically significant?
- Page 6, line 58: Does the 17.1% include daily or ever users of e-cigarettes? And does this include both e-cigarettes with and without nicotine?
- Page 7, line 9-15: No p, beta and CI values are reported in the text whereas they are reported in the abstract. Is that a conscious choice?

Discussion

- The structure of this section can be improved by adding subheading (‘Main findings’, ‘Strengths and limitations’, ‘Implications of findings’, etc.)

	- Repeat the aim of the study at the start of the discussion - Page 8, line 52-54: "The findings are in line with previous studies...." This statement perhaps applies to NRT but it doesn't necessarily apply to e-cigarettes; a recent RCT published by Bullen et al. used a pragmatic approach which suggest that the findings may also apply to the 'real-world' setting. Please make sure this is clear to the reader. Title - The title implies that there is an association found. Perhaps consider to change to title along the lines of: "E-cigarette and nicotine replacement therapy use among smokers not associated with a decline in cigarette consumption: Results of a population-based study using time-series analysis"
--	---

REVIEWER	Lin Zhang University of Minnesota, United States
REVIEW RETURNED	17-May-2017

GENERAL COMMENTS	The authors conducted a time-series analysis to test the association between e-cigarette and nicotine replacement therapy use and cigarette consumption using the ARIMA model. Questions:  1. They are a lot missing data. The authors generally impute the missing data with a fixed value which reflects their best guess of the missing data. However, such imputation ignores the uncertainty in the imputed data. In particular, since there seems to be a big amount of missing data, imputation is likely to largely impact on the analysis result. I wonder if the authors can conduct a sensitivity analysis to examine if the analysis results are robust to the current imputation approach. 2. Page 5, line 45: Can the authors briefly describe how they weighted the data in the preprocessing? 3. The authors used the ARIMAX model for the time-series data analysis, which depends on the parameters including the order and degree of differencing of the moving average and autoregressive model. The authors mentioned that they used ACF and partial ACF plots to determine the values of these parameters. I think this is somewhat ad hoc. I would suggest a more model-based method. That is, try different parameter values and choose the value that best fits the data (using AIC or some other criteria). It would also be good if the authors can provide the exact ARIMAX model that was used for the data analysis (maybe with some exploratory ACF plots). 4. Page 5, line 18: Since the data were log-transformed, I think the coefficient should be interpreted in the log-scale correspondingly. 5. Bayes factors (BFs) are usually used for hypothesis testing for Bayesian models. Can the authors justify their use of BF in this frequentist analysis? Also, it is not clear what priors the authors used (both for the null and alternative hypotheses) in the BF calculation. 6. Page 5, line 11: Statistically, sensitivity analysis is usually conducted to check the robustness of the data analysis to certain assumptions. I am not very clear about the purpose of the sensitivity analyses in this paper. They look to me more like a separate analysis on a different population. Can the authors please clarify on this?
--

VERSION 1 – AUTHOR RESPONSE

Reviewer 1

1. Abstract - Design: only the data collection procedures are described. Please add that it concerns a population-based or ecological study design.

Response: We have now changed this to “Repeated representative cross-sectional population surveys of adults aged 16+ in England”

2. Abstract - Conclusion: This conclusion is only based on the findings in this study. Please nuance the conclusion by stating: “Based on this study ...” and at a term such as “may” or “is likely to”.

Response: We believe that conclusions in abstract should only be based upon findings in the study and we would prefer to avoid reiterating the self-evident point (e.g., <http://www.addictionjournal.org/pages/writing-the-abstract>). Instead, we have made the abstract more directly related to the findings specifically reported in the paper and have added the term ‘likely’: “If use of e-cigarettes and licensed nicotine replacement therapy while smoking acted to reduce cigarette consumption in England between 2006 and 2016, the effect was likely very small at a population level.”

3. Introduction - Page 3, line 17: Which randomised controlled trials are you referring to, please add references.

Response: We have added additional references

4. Introduction - Page 3, line 17: What is meant by ‘non-tobacco nicotine containing products’. Please add some examples (i.e. such as....)

Response: We have added the example Nicotine Replacement Therapy

5. Introduction - Page 3, line 28: What do you mean with ‘licensed products’? This may differ from one country to the other. Please specify.

Response: We have changed this to NRT which is licensed for smoking cessation and harm reduction in England

6. Introduction - Page 3, line 30: There’s a distinction between e-cigarettes with and without nicotine. Please specify here that concerns e-cigarette with nicotine.

Response: We have added “which contain nicotine”

7. Introduction - Page 3, line 31: Add the Cochrane Review on the effectiveness of e-cigarettes for smoking abstinence

Response: We already reference the Cochrane review – reference 15

8. Introduction - Page 3, line 32/33: “This, e-cigarettes may be a more effective aid...”; the Cochrane Review concludes that it’s not yet possible to determine if e-cigarettes are more effective than NRT in helping people to quit. Please add this to this section.

Response: We have changed this sentence to “Thus, although further studies are needed it is possible that e-cigarettes may be a more effective aid for smoking reduction than licensed nicotine products”.

9. Introduction - Page 3, line 37-49 & page 4 line 3-7: This section describes the methods which are used to examine the hypotheses. This should not be described. Please focus on describing the research question and hypotheses in the last paragraph of the introduction and describe the methods in the methods section.

Response: We have edited this section and removed references to the population survey from which the data come but feel that it important to introduce the concept of time series analysis.

10. Methods - Start methods section with a paragraph describing the design of the study.

Response: We have made this amendment.

11. Methods - Overall the writing of this section can be improved to make it more concise and readable (e.g. page 4, line 28-30 “Participants who reported daily or non-daily smoking were asked the following questions”)

Response: We have made this amendment and others to make this section more concise.

12. Methods - Page 4, line 35: Did the survey make a distinction made between e-cigarettes with a without nicotine? If not, please describe the implication of this in the discussion section.

Response: Participants were only asked if they used e-cigarettes, not whether they contained nicotine. We have added the following to the limitations in the discussion “Fifthly, participants were not asked questions regarding potentially important features of the e-cigarette (e.g., nicotine content, flavouring, device type) or frequency and duration of use. It is likely that these factors may play a role in their effectiveness and should be considered in future studies”.

13. Methods - Page 4, line 46-54: the authors describe how smokers were categorised into e-cigarette users and NRT users. What happened when smokers uses both NRT and e-cigarettes?

Response: As this was a population level analysis using aggregated data those who used both e-cigarettes and NRT would have been classed as both NRT and e-cigarettes users i.e. included in both analyses. We believe the description was slightly unclear so have amended this section.

14. Methods - Page 4, line 46-54: perhaps the authors could add definitions of “smoking reduction”, “harm reduction” and “temporary abstinence” of clarification

Response: We already include the definition of harm reduction in the introduction and have now included the definition of smoking reduction and temporary abstinence “NRT for smoking reduction (i.e. during attempts to cut down) or during periods of temporary abstinence (i.e. during periods of time when one is unable to smoke)”

15. Methods - Page 4, line 53: Why was NRT use for temporary abstinence not recorded between Nov-2006 and Jan-2007?

Response: This was due to funding constraint and so the questions were removed for a few waves.

16. Methods - Page 4, line 53-57 & page 5, line 3-14: A lot of imputations, estimations and assumptions are described and were applied to the data. Can the authors explain why they have chosen for this approach and why they haven't decided to focus on the available data from 2011 onwards instead? What are the implications of the imputations/estimations and assumptions - please describe this in the discussion section.

Response: These assumptions were specified in the pre-registered analysis plan and are standard for this type of analysis. Time series analysis requires a long series and so it would not have been possible to restrict the analysis to 2011 onwards. We are confident based on previous surveys that prevalence of e-cigarette use before this time was negligible and slightly different values would not have impacted on the analysis. We now include this as a limitation point in the discussion "Finally, as data were not collected on current e-cigarette use prior to April 2011, prevalence was estimated from use during a quit attempt or from previous studies 21 22. This was necessary to ensure that the time series was long enough for an ARIMAX analysis and is an appropriate approach when data are missing completely at random 16 40. As prevalence was low and relatively stable during this period it is unlikely to have impacted on the reported results."

17. Methods - Page 5, line 17-18: "Smokers taking part in the STS..."; How was the average number of cigarettes smoked per day determined for non-daily smokers?

Response: Non-daily smokers are asked how many cigarettes they smoke per week and this is converted to a daily figure, and this is now included in the manuscript.

18. Methods - Page 5, line 35-38: "Price of cigarettes...."; This sentence will not make a lot of sense for someone who is not familiar with time series analysis. Please explain what is this is needed.

Response: We have now included a definition of differencing "Price of cigarettes is correlated 0.99 with time and will thereby be taken into account by use of differencing (i.e. using the differences between consecutive observation rather the observations themselves) to make the series stationary."

19. Methods - Page 5, line 40: "ANALYSIS" should be a subheading instead of a main heading.

Response: We have now changed it to a sub-heading under methods.

20. Methods - Page 5, paragraph analysis: ARIMAX modelling is a complex approach. The authors provide an explanation of the analyses procedures but this is rather complex. Perhaps this section could be simplified, keeping in mind that a lot of readers are not familiar with ARIMAX modelling?

Response: We have attempted to keep the ARIMAX modelling description to minimum with further details in the pre-published analysis plan. The information given is necessary for the ARIMAX procedure to be followed. We have now defined or explained some of the terms.

21. Methods - Add a paragraph regarding the ethical considerations in this study (informed consent, ethical approval, etc.)

Response: We have added the following to the methods section "Ethical approval for the Smoking Toolkit Study was granted by the UCL ethics committee (ID 0498/001)"

22. Results - Perhaps the authors can consider adding a table with descriptive statistics of the main variables (e.g. for each data collection wave the % of daily and non-daily smokers, the % of e-cigarette and NRT users for the different purposes, and the average number of cigarettes smoked).

Response: This table could also include some sociodemographic characteristics of participants (e.g. age, sex, ethnicity, education).

This data are displayed graphically in the figures and can be obtained directly from the lead author as part of the data sharing agreement. As data are aggregated it does not make sense to report changes over time in age, sex, ethnicity and education as these have remained relatively stable.

23. Results - Page 5, paragraph "Main analyses": start with describing the trend in the outcome variable (i.e. average daily cigarette consumption), followed by a description of the trends in the explanatory variables.

Response: We now start by describing the change in cigarette consumption over time.

24. Results - Page 5, paragraph "Main analyses": where the declining trends in cigarette consumption and NRT use and the increasing trend in e-cigarette use statistically significant?

Response: A formal trend analysis of cigarette use, NRT use and e-cigarette use was not the aim of the current study. We have addressed this previously in a study published in THORAX (<http://dx.doi.org/10.1136/thoraxjnl-2015-206801>)

25. Results- Page 6, line 58: Does the 17.1% include daily or ever users of e-cigarettes? And does this include both e-cigarettes with and without nicotine?

Please see responses above. This includes everyone who used e-cigarettes daily or non-daily and does not distinguish between those using nicotine and non-nicotine containing e-cigarettes as this was not assessed.

26. Results - Page 7, line 9-15: No p, beta and CI values are reported in the text whereas they are reported in the abstract. Is that a conscious choice?

Response: They are reported in the tables so we did not wish to repeat them in the text.

27. Discussion - The structure of this section can be improved by adding subheading ('Main findings', 'Strengths and limitations', 'Implications of findings', etc.)

Response: We have now added subheadings to the discussion

28. Discussion - Repeat the aim of the study at the start of the discussion

Response: An overall summary of the objectives of the study is given at the start of the discussion

29. Discussion - Page 8, line 52-54: "The findings are in line with previous studies...." This statement perhaps applies to NRT but it doesn't necessarily apply to e-cigarettes; a recent RCT published by Bullen et al. used a pragmatic approach which suggest that the findings may also apply to the 'real-world' setting. Please make sure this is clear to the reader.

Response: We now include the reference by Bullen and have also added the following based on pragmatic studies at the individual level "The finding that e-cigarette use was not associated with reductions in consumption at a population level is consistent with previous real-world studies at the individual level. These have found little change in consumption among ever e-cigarette users and that only a minority of daily users manage to reduce by a substantial amount which is not likely to be detected at a population level.

The findings of a recent pragmatic controlled trial, whereby 60% of participants using e-cigarettes had managed to reduce by over 50% by 6 months follow-up, suggests that the lack of effectiveness at a population level may not be the consequence of poor behavioural support.”

30. Title - The title implies that there is an association found. Perhaps consider to change to title along the lines of: “E-cigarette and nicotine replacement therapy use among smokers not associated with a decline in cigarette consumption: Results of a population-based study using time-series analysis”

Response: We have changed the title to “Is prevalence of e-cigarette and nicotine replacement therapy use among smokers associated with average cigarette consumption in England? A time-series analysis”

Reviewer 2

1. I think that the authors missed their opportunity to discuss the findings and their possible implications further in the Discussion. For example, what kind of implication these findings provide e.g. for the current public and/or scientific debate around harm reduction. There are many types of NRT products available. Moreover, there are other forms of tobacco use (snus/snuff). These aspects might be worth to consider in the Discussion as well.

Response: We have now included a separate section on the implications of the findings in the discussion. We do not discuss snus/snuff as these products are banned in England.

2. In future studies, supplemental analysis taking into account SES would provide interesting results. There could be greater accumulation of e-cigarettes and NRT products use in high-SES than low-SES smokers.

Response: As data were aggregated to form the time-series it was not possible to adjust for SES but we agree this would be an interesting analysis for individual level data.

3. Introduction (1st pg): References are mainly limited to the papers made by the author in England. If relevant data is published outside England, I strongly recommend citations of other related work as well.

Response: We now reference the systematic review by Moore and colleagues and reference 4 although by the authors is a systematic review of studies from around the world.

4. Finally, I wonder how response pattern and non-participation to these surveys might have changed over time. Authors could briefly touch this issue in the manuscript.

Response: The baseline survey uses a type of random location sampling, which is a hybrid between random probability and simple quota sampling. England is first split into 171,356 ‘Output Areas’, comprising of approximately 300 households. These areas are then randomly allocated to interviewers, who travel to their selected areas and conduct the electronic interviews with one member of the household. This method of sampling is often seen as superior to conventional quota sampling because the choice of properties approached is significantly reduced by randomly allocating small output areas to the interviewers. However, as interviewers still choose the houses within these particular areas, a response rate cannot be calculated. This is because there is no definite gross sample, with units fulfilling the criteria of the quota being interchangeable.

Reviewer 3

1. They are a lot missing data. The authors generally impute the missing data with a fixed value which reflects their best guess of the missing data. However, such imputation ignores the uncertainty in the imputed data. In particular, since there seems to be a big amount of missing data, imputation is likely to largely impact on the analysis result. I wonder if the authors can conduct a sensitivity analysis to examine if the analysis results are robust to the current imputation approach.

Response: The missing data just relate to a time when we can be confident that e-cigarette use was very rare. We are unaware of how to assess this in a sensitivity analysis but now include this as a discussion point "Finally, as data were not collected on current e-cigarette use prior to April 2011, prevalence was estimated from use during a quit attempt or from previous studies 21 22. This was necessary to ensure that the time series was long enough for an ARIMAX analysis and is an appropriate approach when data are missing completely at random 16 40. As prevalence was low and relatively stable during this period it is unlikely to have impacted on the reported results."

2. Page 5, line 45: Can the authors briefly describe how they weighted the data in the preprocessing?

Response: We now include the following in the methods section "Data were weighted prior to the analyse to match the population in England using a rim (marginal) weighting technique. This involves an iterative sequence of weighting adjustments whereby separate nationally representative target profiles are set (for gender, working status, children in the household, age, social-grade and region). This process is then repeated until all variables match the specified targets (for more details see 20)."

3. The authors used the ARIMAX model for the time-series data analysis, which depends on the parameters including the order and degree of differencing of the moving average and autoregressive model. The authors mentioned that they used ACF and partial ACF plots to determine the values of these parameters. I think this is somewhat ad hoc. I would suggest a more model-based method. That is, try different parameter values and choose the value that best fits the data (using AIC or some other criteria). It would also be good if the authors can provide the exact ARIMAX model that was used for the data analysis (maybe with some exploratory ACF plots).

Response: We used the ACF and PCAF to determine the baseline model and also checked this selection using the AIC. Full details of the analysis are given in the pre-published analysis plan. The exact ARIMAX models are given at the bottom of tables 1, 2 and 3.

4. Page 5, line 18: Since the data were log-transformed, I think the coefficient should be interpreted in the log-scale correspondingly.

Response: As both the input and output time series were log transformed they can be interpreted in terms of elasticities i.e. a change of x percent of the mean for the series A is associated with a change of Y percent of the mean for the series B.

5. Bayes factors (BFs) are usually used for hypothesis testing for Bayesian models. Can the authors justify their use of BF in this frequentist analysis? Also, it is not clear what priors the authors used (both for the null and alternative hypotheses) in the BF calculation.

Response: This was not a full Bayesian analysis and only Bayes Factors were calculated assuming a half-normal distribution. This is recommend approach for interpreting null findings from frequentist analyses as they can inform whether the data are insensitive or if there is evidence for the null (see the work of Zoltan Dienes).

6. Page 5, line 11: Statistically, sensitivity analysis is usually conducted to check the robustness of the data analysis to certain assumptions. I am not very clear about the purpose of the sensitivity analyses in this paper. They look to me more like a separate analysis on a different population. Can the authors please clarify on this?

Response: The analyses are on the same population but exclude those who were not using e-cigarettes or NRT daily. These analyses were pre-specified in the published analysis plan. They were sensitivity analyses as we wanted to check whether the main results differed when only those using e-cigarettes and NRT daily were included. It might be hypothesised that daily use would impact more on cigarette consumption.

VERSION 2 – REVIEW

REVIEWER	Lin Zhang University of Minnesota, United States
REVIEW RETURNED	07-Jul-2017
GENERAL COMMENTS	The review questions were well addressed. I have no further questions.